# Magnetic crystalline-symmetry-protected axion electrodynamics and field-tunable unpinned Dirac cones in EuIn$_2$As$_2$

S. X. M. Riberolles [1✉], T. V. Trevisan[1,2], B. Kuthanazhi[1,2], T. W. Heitmann[3], F. Ye [4], D. C. Johnston[1,2], S. L. Bud'ko[1,2], D. H. Ryan[5], P. C. Canfield[1,2], A. Kreyssig[1,2], A. Vishwanath[6], R. J. McQueeney[1,2], L. -L. Wang[1,2], P. P. Orth [1,2] & B. G. Ueland [1✉]

Knowledge of magnetic symmetry is vital for exploiting nontrivial surface states of magnetic topological materials. EuIn$_2$As$_2$ is an excellent example, as it is predicted to have collinear antiferromagnetic order where the magnetic moment direction determines either a topological-crystalline-insulator phase supporting axion electrodynamics or a higher-order-topological-insulator phase with chiral hinge states. Here, we use neutron diffraction, symmetry analysis, and density functional theory results to demonstrate that EuIn$_2$As$_2$ actually exhibits low-symmetry helical antiferromagnetic order which makes it a stoichiometric magnetic topological-crystalline axion insulator protected by the combination of a 180° rotation and time-reversal symmetries: $C_2 \times \mathcal{T} = 2'$. Surfaces protected by $2'$ are expected to have an exotic gapless Dirac cone which is unpinned to specific crystal momenta. All other surfaces have gapped Dirac cones and exhibit half-integer quantum anomalous Hall conductivity. We predict that the direction of a modest applied magnetic field of $\mu_0 H \approx 1$ to 2 T can tune between gapless and gapped surface states.

[1] Ames Laboratory, Ames, IA 50011, USA. [2] Department of Physics and Astronomy, Iowa State University, Ames, IA, USA. [3] University of Missouri Research Reactor, Columbia, MO, USA. [4] Oak Ridge National Laboratory, Oak Ridge, TN, USA. [5] Physics Department and Centre for the Physics of Materials, McGill University, Montreal, QC, Canada. [6] Department of Physics and Astronomy, Harvard University, Cambridge, MA, USA. ✉email: simon.riberolles@gmail.com; bgueland@ameslab.gov

Electrons attaining a nontrivial Berry phase due to symmetry-protected features in the electronic-band structure[1–4] and/or the presence of noncoplanar magnetic order[5–7] can lead to astonishing topological physical properties such as dissipationless chiral-charge transport, quantum anomalous Hall (QAH) effect, and axion electrodynamics[4,8]. Whereas topological-crystalline insulators (TCIs) are broadly defined as insulators with nontrivial topological properties protected by crystalline symmetry, magnetic TCIs offer the possibility of tuning topological properties via manipulating the magnetic order[4]. Indeed, much theoretical effort is focused on predicting magnetic crystalline materials with nontrivial topological states by considering symmetries associated with the magnetic space groups (MSGs) describing their magnetic order[9–11]. A database with predictions for nontrivial topological band structures based on MSG symmetries provides important guidance towards finding new magnetic topological materials using high-throughput ab initio studies[11]. However, an often limiting bottleneck is detailed knowledge of a candidate's intrinsic magnetic order, which is difficult to predict theoretically. Determining such order can be a subtle task requiring significant experimental effort.

QAH and axion insulators (AXIs) are particularly attractive topological states as the former manifests quantized Hall conductivity in the absence of an applied magnetic field, and the latter shares similarities with the axion particle in quantum chromodynamics[12]. AXIs exhibit the topological magnetoelectric effect for which an applied electric field $\mathbf{E}$ induces a parallel magnetization $\mathbf{M}$ or a magnetic field $\mathbf{H}$ induces a parallel electric polarization[4]. An AXI requires that the axion angle $\theta$ in the action of axion electrodynamics $S_\theta = \theta \frac{e^2}{4\pi^2} \int \mathrm{d}t\, \mathrm{d}^3 r \mathbf{E} \cdot \mathbf{B}$ is $\theta = \pi$, which leads to the presence of half-integer QAH-type conductivity on insulating surfaces[10,13]. Here, $\mathbf{B} = \mu_0 (\mathbf{H} + \mathbf{M})$ is the magnetic induction, $\mu_0$ is the permeability of free space, and $e$ is the electron charge. $\theta$ is quantized to zero or $\pi$ in the presence of either time-reversal $\mathcal{T}$ or inversion $\mathcal{I}$ symmetry, but also any other symmetry operation that reverses an odd number of space–time coordinates [14].

Hexagonal EuIn$_2$As$_2$ [space group $P6_3/mmc$ (No. 194) with lattice parameters $a = 4.178(3)$ Å and $c = 17.75(2)$ Å] is a magnetic TCI built of alternating Eu and In$_2$As$_2$ layers stacked along $\mathbf{c}$ as shown in Fig. 1c[15,16]. The magnetic Eu$^{2+}$ (spin $S = \frac{7}{2}$) layers undergo antiferromagnetic (AF) ordering at a Néel temperature of $T_N \approx 18$ K[17] which density functional theory (DFT) calculations predict to be A-type[18]. A-type order is collinear, with the ordered Eu magnetic moments $\boldsymbol{\mu}$ ferromagnetically aligning in each layer and the layers stacking AF along $\mathbf{c}$. Theory predicts that depending on the orientation of $\boldsymbol{\mu}$, the A-type order leads to an AXI that is either a TCI with some gapless surfaces or a higher-order topological insulator with chiral-hinge states[17,18]. Attractively, the ordered Eu moments may influence topological fermions in In$_2$As$_2$ layers, providing a path for in situ control of band topology.

Below we detail our discovery of low-symmetry broken-helix magnetic order in EuIn$_2$As$_2$ using single-crystal neutron diffraction. We find that the broken-helix order has $2'$ symmetry elements along specific crystalline axes that lead to the emergence of an AXI in the absence of $\mathcal{I}$ and $\mathcal{T}$, both of which are broken by the magnetic ordering. The $2'$ symmetry element denotes the product of a two-fold rotation ($C_2$) and the $\mathcal{T}$ operation: $2' = C_2 \times \mathcal{T}$. $C_2$ rotates the lattice and the spins around the axis by $\pi$ and $\mathcal{T}$ then reverses the direction of the spins. As shown in detail in Supplementary Figs. 16 and 17, the $2'$ transformations leave the broken-helix order invariant and protect gapless surface Dirac cones that are not pinned to time-reversal-invariant

momenta (TRIM). Surfaces not associated with a $2'$ axis have gapped Dirac cones and exhibit half-integer QAH-type conductivity. Our symmetry analyses and magnetization data predict that the surface states are highly tunable by a modest field of $\mu_0 H \approx 1$ to 2 T which is strong enough to polarize the magnetic moments along its direction. This induces a gap and, depending on the field's direction, creates new unpinned gapless states on previously gapped surfaces.

## Results

**Neutron diffraction determination of the AF order.** Figure 1a shows neutron diffraction data for the ($h0l$) reciprocal-lattice plane taken below $T_N \approx 18$ K at $T = 6$ K, and Fig. 1b shows data along the ($00l$) direction taken above and below $T_N$ at 30 and 6 K. Nuclear Bragg peaks occur in Fig. 1b at $l = 2n$ positions for both temperatures whereas magnetic Bragg peaks only appear in the 6 K data. Two AF propagation vectors define the locations of the magnetic Bragg peaks: (1) peaks at $l = 2n \pm \tau_{1z}$ positions correspond to $\boldsymbol{\tau}_1 = (0, 0, \tau_{1z})$ with $\tau_{1z} = 0.303(1)$; (2) nuclear-forbidden peaks at $l = 2n + 1$ positions in Fig. 1b correspond to $\boldsymbol{\tau}_2 = (0, 0, 1)$. [$\boldsymbol{\tau}_2 = (0, 0, 1)$ is equivalent to $\boldsymbol{\tau}_2 = (0, 0, 0)$ for $P6_3/mmc$. We use $\boldsymbol{\tau}_2 = (0, 0, 1)$ to facilitate presentation of the diffraction data.] Within the ($h0l$) reciprocal-lattice plane, magnetic Bragg peaks matching $\boldsymbol{\tau}_2$ overlap with nuclear Bragg peaks with odd values of $l$ when the conditions $h = 3n + 1$ or $h = 3n + 2$ ($n =$ integer) are satisfied. The predicted A-type order would be consistent with magnetic Bragg peaks appearing only at positions corresponding to $\boldsymbol{\tau}_2$. However, the existence of additional magnetic Bragg peaks at $\boldsymbol{\tau}_1$ reveals the presence of more complex helical or spin-density-wave type (itinerant) AF order. Our measurements cannot distinguish between these two types of order, but our DFT calculations reveal that the Eu $4f$ bands are located well below the Fermi level which is more supportive of local-moment helical order.

Using magnetic symmetry analysis and single-crystal refinements to our data from the TRIAX triple-axis spectrometer and the CORELLI time-of-flight spectrometer, we discover that EuIn$_2$As$_2$ has the complex broken-helix AF order illustrated in Fig. 1d and Supplementary Fig. 5. It consists of ferromagnetically aligned layers with $\boldsymbol{\mu} \perp \mathbf{c}$ that are helically stacked along $\mathbf{c}$. We find that the orthorhombic MSG $C2'2'2_1$ (No. 20.33) describes the symmetry, but for continuity we refer to directions with respect to the hexagonal chemical unit cell. We find $\mu = 6.0(3)\mu_B$/Eu at $T = 6$ K, which is smaller than the expected value of $7.0\mu_B$/Eu and the saturation moment of $\mu_{sat} = 7.00(6)\mu_B$ shown below and in Supplementary Fig. 11. We note that neutron diffraction studies of other Eu containing compounds report less than $7.0\mu_B$/Eu[19,20], and, as shown below, the ordered moment is still increasing with decreasing temperature below 6 K. Details of the corrections performed to account for the strong thermal neutron absorption of Eu are given in the "Methods" section and Supplementary Note 1.

To analyze the broken-helix order, we assume a tripling of the chemical unit cell since $\tau_z = 0.303(1) \approx \frac{1}{3}$ and the relatively small incommensurability results in maximum disagreement only at the 33rd Eu layer. Referring to Fig. 1d, we designate the two Eu crystallographic sites in $C2'2'2_1$ as red and blue. This in turn indicates red and blue ferromagnetically aligned layers, and allows for defining the helix turn angles $\phi_{rr}$ and $\phi_{rb}$ between successive red–red and red–blue layers, respectively. The MSG symmetry dictates that $\phi_{rr} + 2\phi_{rb} = 180°$ and constrains magnetic moments in the blue layers to lie along $[\bar{1}10]$. Our refinements find that $\phi_{rb} = 127(3)°$ at $T = 6$ K, and more details of the refinements are given in Supplementary Figs. 1–9 and Supplementary Tables 1–3.

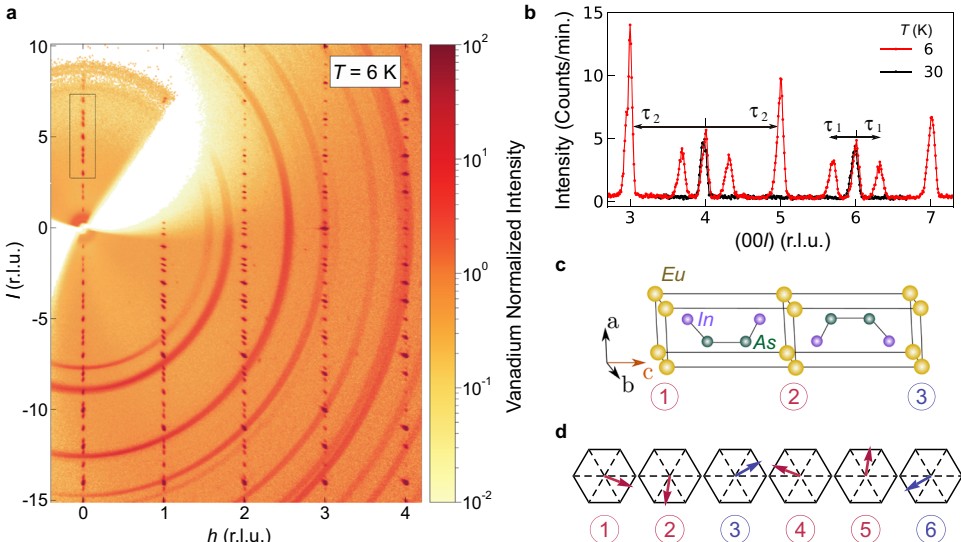

**Fig. 1 Neutron diffraction determination of the broken-helix magnetic order. a** Data for the ($h0l$) reciprocal-lattice plane measured at a temperature of $T = 6$ K with the CORELLI spectrometer[25]. Bragg peaks from the single-crystal sample appear as dots, whereas Bragg peaks due to the polycrystalline Al sample holder appear as rings centered at 0. **b** Scans along the ($00l$) reciprocal-lattice direction for 6 K (red) and 30 K (black) made with the TRIAX instrument. The 30 K data show Bragg peaks solely due to the chemical lattice, and the two sets of magnetic Bragg peaks appearing in the 6 K data are labeled by the antiferromagnetic propagation vectors $\boldsymbol{\tau}_1 = (0, 0, \tau_{1z})$, with $\tau_{1z} = 0.303(1)$, and $\boldsymbol{\tau}_2 = (0, 0, 1)$. [$\boldsymbol{\tau}_2 = (0, 0, 1)$ is equivalent to $\boldsymbol{\tau}_2 = (0, 0, 0)$ for $P6_3/mmc$. We use $\boldsymbol{\tau}_2 = (0, 0, 1)$ to facilitate presentation of the diffraction data.] **c** Chemical unit cell of EuIn$_2$As$_2$ where the numbers correspond to labeling in panel **d**. **d** Layer-by-layer diagram of the 6 K broken-helix order using the notation of the hexagonal chemical unit cell. The magnetic unit cell is actually orthorhombic and is tripled along **c** with the $\mathbf{a}_{\text{ortho}}$ and $\mathbf{b}_{\text{ortho}}$ orthorhombic unit-cell axes lying along $[\bar{1}10]$ and $-[110]$, respectively. Each Eu layer is ferromagnetically aligned with the ordered magnetic moments lying in the **ab** plane. The two symmetry-inequivalent magnetic Eu sites in the magnetic unit cell are colored red (layers 1, 2, 4, and 5) and blue (layers 3 and 6). Moments in the blue layers are constrained by symmetry to lie along $[\bar{1}10]$.

We simulate in Fig. 2a the integrated intensities for representative magnetic Bragg peaks for different values of $\phi_{\text{rb}}$ using the determined MSG and $\boldsymbol{\mu} \perp \mathbf{c}$. Magnetic Bragg peaks corresponding to $\boldsymbol{\tau}_2$ disappear as $\phi_{\text{rb}} \to 60°$ (pure 60°-helix order), whereas magnetic Bragg peaks matching $\boldsymbol{\tau}_1$ disappear as $\phi_{\text{rb}} \to 180°$ (A-type order). We find that the pure 60°-helix order can be described by the higher-symmetry MSG $P6_12'2'$ (No. 178.159) and that the A-type order can be described by MSG $Cm'c'm$ (No. 63.462). The darker shaded area in Fig. 2a indicates the experimental value of $\phi_{\text{rb}}$ at $T = 6$ K.

Figure 2b, c reveals the temperature evolution of the AF order. Figure 2b displays the measured integrated intensities of the $(2, 0, 2) \pm \boldsymbol{\tau}_1$ and $(2, 0, 1)$ magnetic Bragg peaks scaled to 1 at $T = 6$ K. The curves suggest that $\mu$ continues to increase with decreasing temperature below 6 K. Two transitions are evident: the incommensurate magnetic Bragg peaks corresponding to $\boldsymbol{\tau}_1$ with $\tau_{1z} \approx \frac{1}{3}$ emerge at $T_{\text{N1}} = 17.6(2)$ K and commensurate magnetic Bragg peaks corresponding to $\boldsymbol{\tau}_2$ appear at $T_{\text{N2}} = 16.2(1)$ K. These transitions also appear in the magnetic susceptibility, resistance, and $^{151}$Eu Mössbauer data shown in Supplementary Fig. 10. Figure 2c displays the temperature dependence of $\boldsymbol{\tau}_1$ which indicates that pure 60°-helix order first emerges at $T_{\text{N1}}$, and $\tau_{1z}$ decreases upon cooling until $T_{\text{N2}}$ where broken-helix order emerges. This sequence follows the rightmost path in the group-subgroup chart in Supplementary Fig. 18 which traces second-order magnetic transitions from the paramagnetic state to pure 60°-helix order to broken-helix order.

The inset to Fig. 2b shows the evolution of the helix turn angle $\phi$ below $T_{\text{N1}}$. $\phi \approx 60°$ at $T_{\text{N1}}$ and slightly decreases upon cooling for $T_{\text{N2}} < T < T_{\text{N1}}$. The small change from 60° reflects the change in $\tau_{1z}$. For $T < T_{\text{N2}}$, the broken-helix order emerges and the inset to Fig. 2b plots $\phi = \phi_{\text{rb}}$, which is calculated from data shown in Supplementary Fig. 9. $\phi_{\text{rb}}$ rapidly increases immediately below $T_{\text{N2}}$ and approaches $\phi_{\text{rb}} = 130°$ as $T \to 0$ K. The temperature

evolution of $\phi_{\text{rb}}$ below $T_{\text{N2}}$, while maintaining $\tau_{1z} = 0.303(1)$, agrees with the calculations in Fig. 2a showing that broken-helix order persists over a range of $\phi_{\text{rb}}$.

**Density functional theory calculations and symmetry analyses.** Having established and described the emergence of complex helical order, we now discuss and compare the impacts of broken-helix, pure 60°-helix, and A-type order on the electronic-band structure and topology. First, similar to previous reports[18], our DFT calculations and symmetry analyses find that A-type order creates an AXI state with inverted bulk bands near $\Gamma$ and an electronic-band gap of $\approx 100$ meV. Looking at the phase diagram in Fig. 2a, we realize that by adiabatic continuation the magnetic symmetry can be lowered away from the pure 60°-helix or A-type ordered states by varying $\phi_{\text{rb}}$ while maintaining the constraints that $\phi_{\text{rr}} + 2\phi_{\text{rb}} = 180°$ and ordered moments in the blue layers lie along $[\bar{1}10]$. For example, starting from A-type order, reducing $\phi_{\text{rb}}$ from 180° immediately breaks the mirror and $\mathcal{I}$ symmetries present in MSG $Cm'c'm$.

Using this adiabatic continuation approach, we find via DFT calculations that a full band gap persists for any value of $\phi_{\text{rb}}$. Figure 3a shows a diagram of the Brillouin zone and Fig. 3b shows the calculated electronic bands for $\phi_{\text{rb}} = 130°$. Figure 3c focuses on the gap region near $\Gamma$ where band inversion is highlighted by green bars denoting As $4p_z$ orbital character, and Fig. 3d demonstrates that the inverted gap remains for pure 60°-helix order. Supplementary Figure 15 shows that this result holds for other values of $\phi_{\text{rb}}$ spanning $180° \leq \phi_{\text{rb}} \leq 60°$. Thus, the inverted gap remains across all three magnetic orders connected by the internal parameter $\phi_{\text{rb}}$ and indicates that the topological phase is robust to changes to $\phi_{\text{rb}}$. In contrast, we find that bands cross the Fermi level for the case of ferromagnetic order as shown in Supplementary Fig. 15. This opens the possibility of controlling the band structure by applying a magnetic field strong enough to

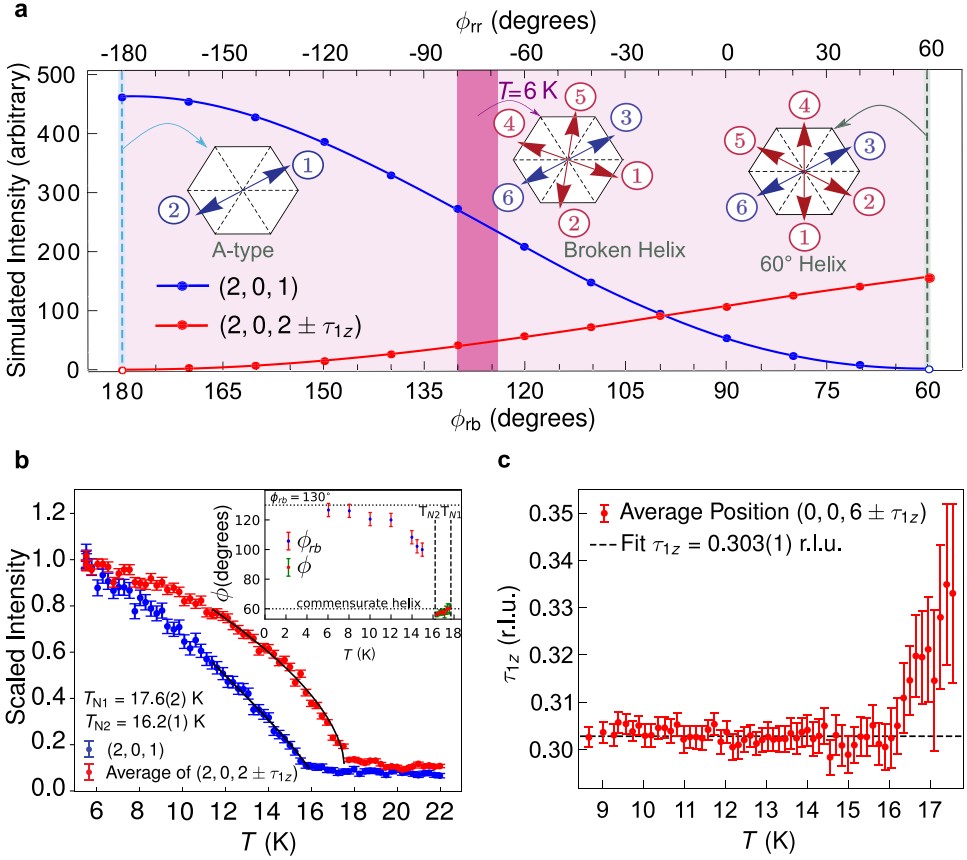

**Fig. 2 Details of the broken-helix magnetic order. a** Simulations of the magnetic neutron diffraction intensity for the $(2, 0, 1)$ Bragg peak and the average of the $(2, 0, 2) \pm \tau_1$ Bragg peaks as functions of the broken-helix turn angles $\phi_{rb}$ and $\phi_{rr}$. **b** Temperature dependencies of the integrated intensity of the $(2, 0, 1)$ Bragg peak and the average integrated intensity for the $(2, 0, 2) \pm \tau_1$ magnetic Bragg peaks measured on the CORELLI spectrometer and scaled to 1 at $T = 6$ K. The inset displays the helix turn angles $\phi$ in the pure 60°-helix phase ($T_{N2} < T \leq T_{N1}$) and $\phi_{rb}$ for the broken-helix phase ($T \leq T_{N2}$). **c** Temperature dependence of $\tau_1 = (0, 0, \tau_{1z})$ from the average of the positions of the $(0, 0, 6) \pm \tau_1$ magnetic Bragg peaks. In **b**, $\phi(T)$ is calculated from the temperature evolution of $\tau_{1z}$. $\phi_{rb}(T)$ is calculated from the temperature dependence of the ratio of the integrated intensity of the $(0, 0, 5)$ magnetic Bragg peak to that of the $(006) - \tau_1$ magnetic Bragg peak. These data are from measurements made on the TRIAX spectrometer and are shown in Supplementary Fig. 9.

align $\mu$ along a single direction. This point is discussed further below.

## Discussion

With these results in hand, we now address why $EuIn_2As_2$ is an AXI in its magnetically-ordered phases despite the absence of $\mathcal{I}$. We determine $\theta = \pi$ in $S_\theta$ for the broken-helix order by finding that $\theta = \pi$ in the presence of $\mathcal{I}$ for $\phi_{rb} = 180°$ (A-type order) via calculating the parity-based symmetry indicator $\mathbb{Z}_4 = \frac{1}{2} \sum_{k \in \text{TRIMs}} (n_k^+ - n_k^-) \mod 4 = 2$ [10,11,18,21]. Here, $n_k^\pm \geq 0$ denotes the total number of filled bands with parity eigenvalue $\pm 1$ at TRIM $k$. Since the band gap remains for $60° \leq \phi_{rb} \leq 180°$, adiabatic continuity ensures that $\theta = \pi$ for all three magnetic structures, and, remarkably, the entire family of A-type, broken-, and pure 60°-helix states are AXIs as long as the Fermi energy resides in the band gap. It is the $2'$ symmetry operator along [110] and $[\bar{1}10]$ for the broken-helix order and along [110] and [100] for the pure 60°-helix order that permits an AXI in the absence of $\mathcal{I}$ [14].

The presence of $2'$ symmetry axes for the broken-helix order enforces an odd number of gapless unpinned surface Dirac cones on the $(\bar{1}10)$ and $(110)$ surfaces. More specifically, the unpinned gapless Dirac cones occur on surfaces perpendicular to the $2'$ axes. These exotic topological surface states appear in the presence of $2'$ because it reverses an odd number of space–time

coordinates [14]. Figure 3e, f shows the surface band structure for $(110)$ wherein a gapless Dirac cone is seen which is offset from $\bar{\Gamma}$ along $\bar{\Gamma}$–$\bar{X}$. This result contrasts with the gapped Dirac cone at $\bar{\Gamma}$ on the $(001)$ surface shown in Fig. 3g. For the case of no magnetic order, Fig. 3h shows a gapless Dirac cone on the $(001)$ surface pinned to $\bar{\Gamma}$. These calculations did not include the Eu $4f$ orbitals. We find that $\mathbb{Z}_2 = 1$, which signals a strong topological insulator, rather than an AXI phase, for the non-magnetic state.

The AXI state protected by $2'$ symmetry is different from the AXI phase protected by a combination of $\mathcal{T}$ and a translation recently proposed for $MnBi_2Te_4$ (ref. [22]). In $MnBi_2Te_4$, this symmetry is only unbroken on surfaces that contain the translation vector, where it leads to gapless Dirac cones that are pinned to TRIM. The Dirac cones are gapped on all other surfaces. In contrast, in $EuIn_2As_2$ the AXI state is protected by $2'$ symmetry, leading to unpinned Dirac surface states whenever the $2'$ axis is normal to the surface and gapped surface states appear on all other surfaces.

The different topological surface states resulting from the presence or absence of $2'$ symmetry leads to differing physical properties. To facilitate this discussion, Fig. 4a shows the relationships between real-space and reciprocal-lattice directions, and the direction(s) of the $2'$ axis/axes are listed in Fig. 4c for various magnetic states. For the broken-helix order, $(001)$ and other gapped surfaces without a $2'$ axis exhibit half-integer QAH-type conductivity [4]. On the other hand, similar to a 3D topological

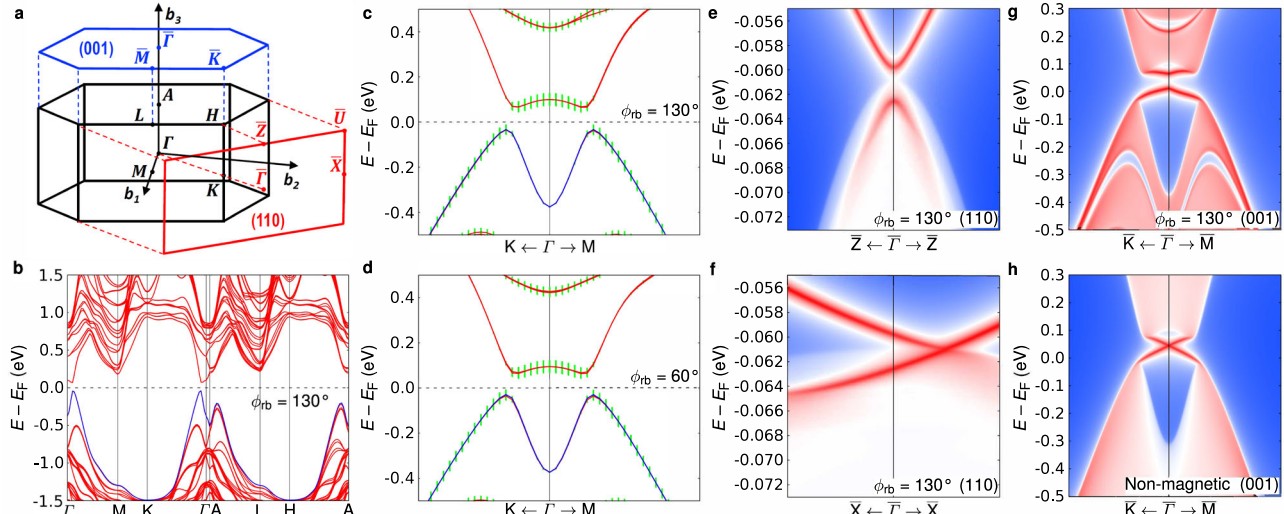

**Fig. 3 Bulk and surface electronic-band structures for various magnetic ground states. a** Hexagonal Brillouin zone and the projected (110) and (001) surface-Brillouin zones with high-symmetry points (capital letters) and reciprocal-lattice axes ($\mathbf{b}_1 = \mathbf{a}^\star$, $\mathbf{b}_2 = \mathbf{b}^\star$, and $\mathbf{b}_3 = \mathbf{c}^\star$) indicated. **b** Bulk band structure along high-symmetry paths for broken-helix order with $\phi_{rb} = 130°$. $E_F$ is the Fermi energy. **c, d** Views of the minimal gap region along the K-$\Gamma$-M direction for $\phi_{rb} = 130°$ (broken-helix order) (**c**) and $\phi_{rb} = 60°$ (pure 60°-helix order) (**d**) . The top valence band according to simple band filling is indicated in blue and the rest of the bands are colored red. Green vertical bars show As $4p_z$ orbital character and indicate band inversion near the minimal gap. **e–h** Surface bands for the (110) (**e, f**) and (001) (**g, h**) projected surfaces along high-symmetry lines for $\phi_{rb} = 130°$ (**e–g**). A Dirac cone exists on each surface, but the presence of a gap and whether or not the cone is pinned to a time-reversal-invariant momentum (TRIM) depends on symmetry: a gapless Dirac cone unpinned from a TRIM occurs for the (110) surface preserving the $2'$ symmetry axis. On the other hand, the (001) surface (**g**) has a gapped Dirac cone at $\bar{\Gamma}$ due to the absence of a $2'$ axis and any other symmetry that reverses an odd number of space–time coordinates. The surfaces with gapped Dirac cones support half-integer quantum-anomalous-Hall-type conductivity. The non-magnetic calculation for the (001) surface (**h**) has a gapless Dirac cone pinned to $\bar{\Gamma}$. Note that panels **e** and **f** are zoomed in around the bottom of the band gap to show the details of the surface Dirac cone. $E_F$ is inside the gap and the surface Dirac point is at $E_F - 0.061$ eV, very close to the top of the valence band projection. More details of the surface band structure are found in Supplementary Fig. 15g.

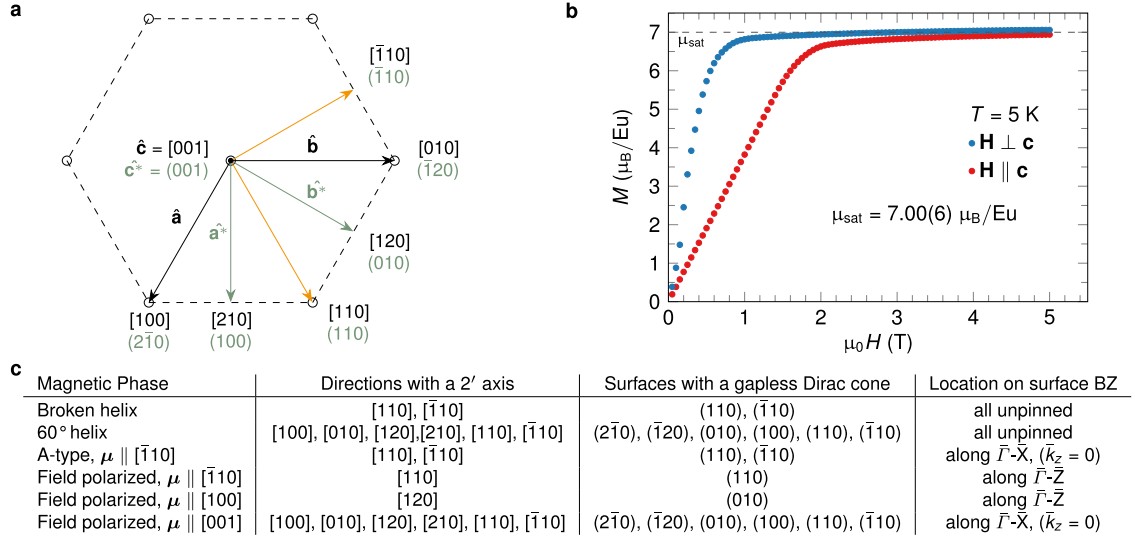

**Fig. 4 Tuning surface Dirac cones via an applied magnetic field. a** Diagram showing real-space (denoted by brackets) and reciprocal-space (denoted by parentheses) directions for the **ab** plane. The magnetic orthorhombic unit cells for broken-helix and A-type orders discussed in the main text have their $\mathbf{a}_{orth}$ and $\mathbf{b}_{orth}$ axes along $[\bar{1}10]$ and $-[110]$, respectively. **b** Magnetization versus magnetic field curves for a temperature of $T = 5$ K and the field applied either perpendicular or parallel to **c**. **c** A table indicating the $2'$ axis/axes, the surface(s) containing a gapless Dirac cone, and the location of the gapless Dirac cone on the projected surface-Brillouin Zone(s) (BZ) for various zero-field and field-polarized magnetic states. The direction of the ordered magnetic moment $\boldsymbol{\mu}$ is indicated where appropriate. The results for the broken-helix and 60°-helix phases assume that the blue Eu layers lie along $[\bar{1}10]$.

insulator, the gapless surface Dirac cones on (110) and ($\bar{1}$10) support dissipationless charge transport where sweeping $E_F$ through the Dirac point switches the handedness of the chirality associated with the itinerant charge. Similar analyses of the surface states can be performed for the pure 60°-helix and A-type order phases. In particular, Fig. 4c shows that the pure 60°-helix phase supports multiple surfaces with an unpinned gapless Dirac cone. The consequences of the gapless Dirac cones being exotically unpinned from TRIM, which require absence of Kramer's degeneracy, however, lacks investigation.

Figure 4c also illustrates that the surface states are highly tunable by an external magnetic field, and Fig. 4b shows that modest field strengths of $\mu_0 H \approx 1$–2 T are sufficient to polarize the Eu magnetic moments along any direction at $T = 5$ K. For example, the row in Fig. 4c corresponding to broken-helix order indicates that a gapless completely unpinned Dirac cone exists only on the $(\bar{1}10)$ and $(110)$ surfaces, and, therefore, all of the other surfaces have a gapped Dirac cone. For a field-polarized state with $\boldsymbol{\mu}\|[100]$, Fig. 4c indicates that only the (010) surface has a gapless surface Dirac cone and its node is constrained to lie along $\bar{\Gamma}$-$\bar{Z}$. Thus, a field provides an effective means for switching by inducing a gap in previously gapless surface states while eliminating the gap in previously gapped surface states. Further, the position of the node of the gapless Dirac cone can change from being completely unpinned to being constrained to a line. Figure 4c also highlights the direct correlation between the existence of a $2'$ axis and gapless surface states. More details of our symmetry analyses are given in Supplementary Table 4 and in the associated discussion in Supplementary Note 5.

Finally, a true AXI state only occurs for insulating compounds, but resistance and angle-resolved-electron-photoemission data for EuIn2As2 are consistent with it being a slightly hole-doped compensated semimetal[17,23], see Supplementary Fig. 10c. Thus, $E_F$ needs to be shifted into the bulk gap for an AXI to be observed. Fortunately, the robust quantization of $\theta = \pi$ with respect to changes of $\phi_{rb}$ suggests that small perturbations to the magnetic order potentially caused via tuning $E_F$ will not destroy the symmetry leading to an AXI state. Further, the fact that $\boldsymbol{\tau}_1$ is not exactly $\frac{1}{3}$ does not invalidate our analysis, because the slight incommensurability should enter into the calculation of $\theta$ as only a small perturbation. Investigations into growing thin films may permit gating or strain to shift the chemical potential, and growth of films on magnetic substrates may allow for simultaneous gating while facilitating magnetic-field control of the topological surface states.

To conclude, we have established that complex helical magnetic order emerges in EuIn2As2 upon cooling below $T_{N1} = 17.6(2)$ K (pure 60°-helix order) and $T_{N2} = 16.2(1)$ K (broken-helix order). Both of these helical orders create $2'$ symmetry axes that protect an AXI state in the absence of $\mathcal{I}$. Our DFT calculations and symmetry analyses show that the band topology is robust to changes of the helical turn angles, such that the magnetic order can change from A-type, to broken-helix, to pure 60°-helix order while maintaining the same topological phase. We further find that the existence of $2'$ symmetry axes protects gapless Dirac cones on specific surfaces which are unpinned from TRIM and that the other surfaces possess gapped Dirac cones and exhibit QAH-type conductivity. Our magnetization results show that only a moderate magnetic field of $\mu_0 H \approx 1$–2 T is necessary for tuning the direction of the magnetic moment, which our symmetry analyses show can choose the surfaces supporting gapless unpinned Dirac cones. This opens up the possibility for novel device development as well as further research into the physical properties associated with unpinned surface Dirac cones. For example, experiments exploring if these exotic surface states can be moved in the surface-Brillouin zone via pressure or strain may yield insight into basic material interactions and reveal unexpected properties. The intrinsic low-symmetry broken-helix magnetic order of EuIn2As2 offers an excellent opportunity to study and tune exotic surface states in a magnetic TCI.

## Methods

**Sample synthesis**. Single crystals of EuIn2As2 were grown using a flux method similar to ref. [15] and found to be single phase via powder X-ray diffraction. An initial composition of Eu:In:As = 3:36:9 was weighed out and packed in fritted alumina crucibles[24], followed by sealing in a fused silica tube. The prepared ampoule was first heated up to 300 °C over 2 h and held there for an hour, then heated up to 580 °C over 3 h and held there for 2 h, and finally heated up to 900 °C over 10 h followed by a 2-h dwell. The intermediate dwells were to ensure maximum dissolving and incorporation of the volatile elements into the melt. The final dwell was followed by a 48-h cool down to 770 °C, at which point excess flux was decanted using a centrifuge. This process yielded plate-like crystals of EuIn2As2 with masses of a few milligrams.

**Magnetization experiments**. Magnetization measurements were made on single-crystal samples down to $T = 2$ K under applied magnetic fields up to $\mu_0 H = 5$ T using a Quantum Design, Inc. Superconducting Quantum Interference Device.

**Neutron diffraction experiments**. Neutron diffraction measurements were made on the TRIAX triple-axis spectrometer at the University of Missouri Research Reactor, and on the time-of-flight (TOF) CORELLI spectrometer at the Spallation Neutron Source at Oak Ridge National laboratory[25]. Single-crystal samples with masses of 4.5 and 11.6 mg were used on TRIAX and CORELLI, respectively. The samples were attached to Al sample mounts and cooled down to $T = 6$ K using closed-cycle He refrigerators. The integrated intensities of recorded Bragg peaks were corrected for the effects of neutron absorption using the MAG2POL[26] software according to the procedure described in ref. [27]. The samples were flat rectangular plates with thicknesses of 0.1 mm along [001]. The TRIAX sample has an approximate width of 3 mm along [100] and approximate length of 2 mm along [010]. The CORELLI sample has an approximate width of 3 mm along [110] and approximate length of 3 mm along [$\bar{1}$10].

The TRIAX experiments were performed in elastic mode with incident and final neutron energies of $E_{i,f} = 14.7$ and 30.5 meV. Söller slit collimators with divergences of 60'-60'-80'-80' were inserted before the pyrolytic graphite (PG) (002) monochromator, between the monochromator and sample, between the sample and PG (002) analyzer, and between the analyzer and detector, respectively. PG filters were inserted before and after the sample to eliminate higher-order beam harmonics. The sample's (h0l) reciprocal-lattice plane was aligned within the horizontal scattering plane. In order to collect data for purely nuclear and magnetic Bragg peaks, a series of (h0l) reflections were recorded at $T = 30$ and 6 K, i.e. significantly above $T_{N1}$ and below $T_{N2}$, respectively. Each Bragg peak was fitted with a Gaussian lineshape in order to determine its integrated intensity (area), center, and full-width at half-maximum. The magnetic signals of the low-temperature $\boldsymbol{\tau}_2$ Bragg peaks were then extracted by subtracting the integrated intensity of the 30 K nuclear contribution when appropriate. A Lorentz factor ($L = \frac{1}{\sin 2\theta}$) was also applied. Refinements were made with FULLPROF[28]. Throughout this work, error bars represent one standard deviation.

CORELLI experiments were performed at several temperatures above and below $T_{N1}$ and $T_{N2}$ with the (h0l) plane set horizontal. However, the spectrometer employs 2-D position-sensitive detectors which allow for recording some Bragg peaks above and below this plane. The TOF measurement technique employs a white neutron beam, and the sample was rotated around its vertical axis at discrete intervals in order to cover a large swath of reciprocal space. Individual Bragg peaks were obtained by performing cuts through the 2-D maps recorded by the instrument, returning intensity versus $\mathbf{Q}$ profiles of the peaks. Integrated intensities were in turn calculated by adding up all of the data points located in the peak profile that were above background. To account for the pulsed beam's non-linear distribution of wavelengths, a normalization to vanadium scattering was applied. CORELLI data used for the structural and magnetic refinements presented in Supplementary Note 1 are restricted to a bandwidth of incoming neutron energies of 45–55 meV in order to accurately correct for neutron absorption.

**Electronic-band structure calculations**. Band structures using DFT with spin–orbit coupling were calculated with the Perdew–Burke–Ernzerhof exchange-correlation functional, a plane-wave basis set, and the projected-augmented-wave method as implemented in VASP[29,30]. To account for the half-filled strongly-localized Eu 4f orbitals, a Hubbard-like[31] U value of 5.0 eV is used. For different helical magnetic structure with $\tau \approx (0, 0, 1/3)$, i.e. the hexagonal unit cell is tripled along the **c**-axis with atoms fixed in their bulk positions. A Monkhorst-Pack $12 \times 12 \times 1$ k-point mesh with a Gaussian smearing of 0.05 eV including the $\Gamma$ point and a kinetic-energy cutoff of 250 eV have been used. To search for possible band gap closing points in the full Brillouin Zone, a tight-binding model based on maximally localized Wannier functions[32] was constructed to reproduce closely the bulk band structure including spin–orbit coupling in the range of $E_F \pm 1$ eV with Eu sdf, In sp, and As p orbitals as implemented in WannierTools[33].

## Data availability

The neutron scattering data that support our analysis of the ground state magnetic order in the EuIn2As2 are displayed in the Supplementary Note 1 and/or available in the MDF Open data repository with the identifiers https://doi.org/10.18126/u3j8-aplr and https://doi.org/10.18126/u3j8-aplr.

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

## Acknowledgements

This research was supported by the Center for Advancement of Topological Semimetals, an Energy Frontier Research Center funded by the U.S. Department of Energy Office of Science, Office of Basic Energy Sciences, through the Ames Laboratory under Contract No. DE-AC02-07CH11358. D.C.J., S.L.B., and A.K. were supported by U.S. Department of Energy Office of Science, Office of Basic Energy Sciences, Field Work Proposals at the Ames Laboratory operated under the same contract number. This research used resources at the Spallation Neutron Source, a DOE Office of Science User Facility operated by the Oak Ridge National Laboratory. This research used resources at the Missouri University Research Reactor. Financial support for this work was provided by Fonds Québécois de la Recherche sur la Nature et les Technologies. Much of this work was carried out while D.H.R. was on sabbatical at Iowa State University and Ames Laboratory, and their generous support under the U.S. Department of Energy Office of Science, Office of Basic Energy Sciences Contract No. DE-AC02-07CH11358 during this visit is gratefully acknowledged.

## Author contributions

S.X.M.R., B.G.U., R.J.M., T.W.H., and F.Y. conducted neutron scattering experiments. S.X.M.R. refined the neutron diffraction data and determined the magnetic structure with input from A.K., B.G.U., D.C.J., and R.J.M. B.K., S.L.B., and P.C.C. synthesized crystals and performed magnetization and resistance measurements. D.H.R. performed Mössbauer measurements. T.V.T., P.P.O., L.L.W., and A.V. performed symmetry analyses concerning the topologically protected properties. L.L.W. performed electronic-band-structure calculations.

## Competing interests

The authors declare no competing interests.
