## [Peer Review File · Nature Communications]

Parts of this Peer Review File have been redacted as indicated as we could not obtain permission to publish the reports of reviewer #2.

REVIEWER COMMENTS

Reviewer #1 (Remarks to the Author):

The manuscript addresses the magnetic properties of the hexagonal system EuIn_2As_2 and how the topological properties emerge from these properties. Central to being able to understand the properties is a complete determination of the magnetic structures by magnetic neutron diffraction. Experimentally there are two magnetic phase transitions, and the magnetic structures are more complex than the simple A-type magnetic structure that was originally expected. The overall experimental behavior is further illuminated and confirmed in detail by bulk susceptibility, magnetization, resistivity, and Mossbauer data. Detailed symmetry analysis and DFT calculations were then carried out to elucidate the magnetism and directly relate it to the topological properties. These emergent topological properties are thoroughly analyzed and it is explained in great detail how they can be manipulated with an applied magnetic field. The manuscript is well organized and well written, and therefore it is a pleasure to recommend it for publication. There are a few minor corrections and suggested clarifications as follows.

Fig. 1b. The choice of the two colors, with both filled points, make it is somewhat difficult to distinguish the data for the two temperatures.

The ordered moment size obtained with the neutron diffraction measurements of only 5.6 Bohr magnetons is unexpected given that the full moment of 7 is easily induced in the bulk magnetization measurements with a modest magnetic field. A similar field dependent measurement with neutrons could determine if this discrepancy is actually due to incorrect absorption corrections because of the Eu, or rather might suggest there is a new set of magnetic Bragg peaks that have yet to be discovered. Is that a possibility?

Page 7. You claim via DFT calculations that there is an electronic gap of 100 microvolts. Don't you mean 100 meV?

Fig S7. The placement of the legend in Fig. 7 is unintentionally misleading, with at first glance making it appear as if there is a dramatic jump in the intensity of the incommensurate phase at base temperature, which initially is confirmed in the upper inset where the ratio is near unity, due again to an inappropriate placement of the legend.

Fig. S9a,b. Everywhere in the manuscript the (correct) unit of Tesla is used, except in figure 9 where the units of H in Oe or kOe is used. It is suggested this be changed to Tesla.

Reviewer #2 (Remarks to the Author):

[Redacted]

Reviewer #3 (Remarks to the Author):

This manuscript, by Riberolles et al, reports magnetic crystalline-symmetry-protected axion electrodynamics and field-tunable unpinned Dirac cones in EuIn_2As_2 . In this work, the authors found EuIn_2As_2 is magnetic topological-crystalline axion insulator protected by the $C_2 \times T$ symmetry with a low-symmetry helical antiferromagnetic order. The authors predicted unpinned gapless Dirac cone which could be tuned by applied magnetic field.

Different from the previous reports, this work employed neutron diffraction to demonstrate a new and different magnetic structure. Also the authors proposed a new symmetry $C_2 \times T$ of the combination of C_2 rotation and time-reversal symmetry T for the axion insulator. In this view point, this work is quite interesting and important. But before recommending publishing, I have a few comments for the

authors.

1) The symmetry of $C2xT$. In the manuscript, it is hard to understand this symmetry in the current version for readers, though I have understood it. I suggest the authors to give more detailed description for this symmetry.

2) In fig.3, I do not find any description for the blue line and red line in fig 3b and 3d. In addition, in fig. 3e and 3f, I do not understand why the $E-E_f=0$ does not locate at the energy gap/Dirac cone.

3) For the introduction part. Recently, antiferromagnetic insulator $MnBi_2Te_4$ was recently proposed and confirmed to be axion insulator protected by S symmetry of the combination of transition symmetry and time-reversal symmetry [e.g. PRL 122, 206401 (2019)]. I suggest the authors to have a few discussions on the similarities and differences between $MnBi_2Te_4$ and $EuIn_2As_2$, which may make the readers have a better understanding for the magnetic axion states.

4) For the title "Magnetic crystalline-symmetry-protected axion electrodynamics and field-tunable unpinned Dirac cones in $EuIn_2As_2$ ". What does the axion electrodynamics mean here? How to demonstrate the axion electrodynamics? As my understanding, the axion field in this materials $EuIn_2As_2$ is static/quantized π , not dynamical.

RESPONSE TO REVIEWER COMMENTS

Reviewer #1 (Remarks to the Author):

The manuscript addresses the magnetic properties of the hexagonal system EuIn_2As_2 and how the topological properties emerge from these properties. Central to being able to understand the properties is a complete determination of the magnetic structures by magnetic neutron diffraction. Experimentally there are two magnetic phase transitions, and the magnetic structures are more complex than the simple A-type magnetic structure that was originally expected. The overall experimental behavior is further illuminated and confirmed in detail by bulk susceptibility, magnetization, resistivity, and Mossbauer data. Detailed symmetry analysis and DFT calculations were then carried out to elucidate the magnetism and directly relate it to the topological properties. These emergent topological properties are thoroughly analyzed and it is explained in great detail how they can be manipulated with an applied magnetic field. The manuscript is well organized and well written, and therefore it is a pleasure to recommend it for publication. There are a few minor corrections and suggested clarifications as follows.

Fig. 1b. The choice of the two colors, with both filled points, make it is somewhat difficult to distinguish the data for the two temperatures.

We changed the color choice for this figure in the revised manuscript.

The ordered moment size obtained with the neutron diffraction measurements of only 5.6 Bohr magnetons is unexpected given that the full moment of 7 is easily induced in the bulk magnetization measurements with a modest magnetic field. A similar field dependent measurement with neutrons could determine if this discrepancy is actually due to incorrect absorption corrections because of the Eu, or rather might suggest there is a new set of magnetic Bragg peaks that have yet to be discovered. Is that a possibility?

We think it is unlikely that there is an undiscovered set of magnetic Bragg peaks in the temperature range we studied, as our measurements covered large swaths of reciprocal space. We note that neutron diffraction results obtaining less than the expected ordered moment of $7 \mu_B/\text{Eu}$ in Eu^{2+} containing materials have been previously reported. A couple of examples are now included in the manuscript starting at the bottom of p. 4 (Refs. 23 and 24). The occurrence of $< 7 \mu_B/\text{Eu}$ as $T \sim 0$ K has been explained in quasi-1D EuAs_3 magnet, for example, as being due to the presence of zero point spin deviations, a phenomenon that would be enhanced in a quasi-2D magnet such as EuIn_2As_2 . However, we also now explicitly point out in the manuscript that the order parameter plots (Fig. 2b) indicate that the ordered moment is still increasing with decreasing temperature below 6 K. Finally, as we explain in our response to Referee 2's comment 2, we have also improved the determination of the ordered moment by incorporating refinements of the CORELLI data. Based on the refinements to the TRIAX and CORELLI data we now report $6.0(3) \mu_B/\text{Eu}$ and $\varphi_{\text{rb}} = 127(3)^\circ$.

The suggestion to perform experiments in a magnetic field is an intriguing one, and we indeed are planning follow-on in-field experiments. However, the use of a magnet introduces other issues into the data treatment, since it causes more background scattering and absorption. The range of accessible Q space is also limited by a magnet. As such, it is usual course with single-crystal diffraction to first establish the zero-field structure and then follow changes to the structure with a magnetic field in subsequent experiments. We believe the current manuscript's reporting of the compound's zero-field complex magnetic order and the topological ramifications of such order stand on their own. We further believe this work will also inspire others to investigate the in-field magnetic structures and their topological properties. As pointed out in the table in Fig. 4, there are a number of interesting possibilities for follow-on in-field work.

Page 7. You claim via DFT calculations that there is an electronic gap of 100 microvolts. Don't you mean 100 meV?

We have corrected this typo (now at the center of p. 8).

Fig S7. The placement of the legend in Fig. 7 is unintentionally misleading, with at first glance making it appear as if there is a dramatic jump in the intensity of the incommensurate phase at base temperature, which initially is confirmed in the upper inset where the ratio is near unity, due again to an inappropriate placement of the legend.

The figure (now Supplementary Fig. 9) has been corrected using a better placement of the legends to avoid confusion.

Fig. S9a,b. Everywhere in the manuscript the (correct) unit of Tesla is used, except in figure 9 where the units of H in Oe or kOe is used. It is suggested this be changed to Tesla.

We have changed units in the manuscript to Tesla for magnetic field and now properly refer to the field as $\mu_0 H$ when using units of Tesla.

[Redacted]

Reviewer #3 (Remarks to the Author):

This manuscript, by Riberolles et al, reports magnetic crystalline-symmetry-protected axion electrodynamics and field-tunable unpinned Dirac cones in EuIn_2As_2 . In this work, the authors found EuIn_2As_2 is magnetic topological-crystalline axion insulator protected by the C_2xT symmetry with a low-symmetry helical antiferromagnetic order. The authors predicted unpinned gapless Dirac cone which could be tuned by applied magnetic field.

Different from the previous reports, this work employed neutron diffraction to demonstrate a new and different magnetic structure. Also the authors proposed a new symmetry $C_2 \times T$ of the combination of C_2 rotation and time-reversal symmetry T for the axion insulator. In this viewpoint, this work is quite interesting and important. But before recommending publishing, I have a few comments for the authors.

1) The symmetry of $C_2 \times T$. In the manuscript, it is hard to understand this symmetry in the current version for readers, though I have understood it. I suggest the authors to give more detailed description for this symmetry.

We appreciate the referee's suggestion. To clarify the role of this symmetry, we have added and modified the following statements in the last paragraph of p. 3 of the manuscript:

"The $2'$ symmetry element denotes the product of a two-fold rotation (C_2) and the \mathcal{T} operation: $2' = C_2 \times \mathcal{T}$. C_2 rotates the lattice and the spins around the axis by π and \mathcal{T} then reverses the direction of the spins. As shown in detail in Supplementary Figs. 16 and 17, the $2'$ transformations leave both the pure- and the broken-helix order invariant and protect gapless surface Dirac cones that are not pinned to time-reversal-invariant momenta (TRIM)."

2) In Fig. 3, I do not find any description for the blue line and red line in fig 3b and 3d. In addition, in Figs. 3e and 3f, I do not understand why the $E - E_F = 0$ does not locate at the energy gap/Dirac cone.

We thank the referee for pointing out the lack of description for the colors used in Figs. 3b and 3d. We have added the sentence "The top valence band according to simple band filling is indicated in blue and the rest of the bands are colored red," in the Figure caption. A comparable sentence was also added to the caption of Supplementary Fig. 15.

For the second part of the comment, E_F is inside the gap and the surface Dirac point is at $E_F - 0.061$ eV, very close to the top of valence band projection. Figs. 3e and 3f are zoomed in around the bottom of the band gap to show the details of the surface Dirac cone. To get a full view of the surface band structure and band gap, we have added a new figure in the supplement (Supplementary Fig. 15g). Descriptions of the added panel are included in the figure caption as well as in the corresponding SI text.

Finally, as we explain on p. 12 of the manuscript, EuIn_2As_2 is slightly hole doped and E_F needs to be shifted into the bulk gap for an AXI to occur.

3) For the introduction part. Recently, antiferromagnetic insulator MnBi_2Te_4 was recently proposed and confirmed to be axion insulator protected by S symmetry of the combination of transition symmetry and time-reversal symmetry [e.g. PRL 122, 206401 (2019)]. I suggest the authors to have a few discussions

on the similarities and differences between MnBi_2Te_4 and EuIn_2As_2 , which may make the readers have a better understanding for the magnetic axion states.

We thank the referee for this suggestion and have added the following paragraph to the Discussion section (p. 10):

The AXI state protected by $2'$ symmetry is different from the AXI phase protected by a combination of \mathcal{T} and a translation recently proposed for MnBi_2Te_4 [25]. In MnBi_2Te_4 , this symmetry is only unbroken on surfaces that contain the translation vector, where it leads to gapless Dirac cones that are pinned to TRIM. The Dirac cones are gapped on all other surfaces. In contrast, in EuIn_2As_2 the AXI state is protected by $2'$ symmetry, leading to unpinned Dirac surface states whenever the $2'$ axis is normal to the surface and gapped surface states on all other surfaces.

4) For the title “Magnetic crystalline-symmetry-protected axion electrodynamics and field-tunable unpinned Dirac cones in EuIn_2As_2 ”. What does the axion electrodynamics mean here? How to demonstrate the axion electrodynamics? As my understanding, the axion field in this materials EuIn_2As_2 is static/quantized π , not dynamical.

Indeed, the axion angle is static in our work and $\theta = \pi$ (a constant) in the material. The value $\theta = \pi$ induces a large magnetoelectric coupling, which is described by the well-known $\mathbf{E} \cdot \mathbf{B}$ term in the action. Since the electric and magnetic fields are dynamical, we refer to the situation as “axion electrodynamics.” We note that this term is common in the field when used to describe a fixed value of $\theta = \pi$. For example, an early and influential paper by Essin, Moore and Vanderbilt [Phys. Rev. Lett. 102, 146805 (2010)] uses this term in the title: “Magnetoelectric Polarizability and Axion Electrodynamics in Crystalline Insulators”.

REVIEWERS' COMMENTS

Reviewer #1 (Remarks to the Author):

I have read the revised manuscript. My original recommendation was that this work was suitable for publication with some revisions suggested. The authors have satisfactorily addressed all corrections and suggestions, and therefore my recommendation now is to publish the manuscript.

Reviewer #2 (Remarks to the Author):

[Redacted]

Reviewer #3 (Remarks to the Author):

The authors answered my questions and comments very well. They modified their manuscript following my suggestions. I am satisfied with the response. Now the manuscript seems to reach the standard of Nature Communications.